# Chaos in synthetic microbial communities

**Behzad D. Karkaria**[1], **Angelika Manhart**[2], **Alex J. H. Fedorec**[1], **Chris P. Barnes**[1]*

**1** Department of Cell & Developmental Biology, University College London, London, United Kingdom,
**2** Department of Mathematics, University College London, London, United Kingdom

¤ Current address: Department of Cell & Developmental Biology, University College London, London, United Kingdom

* christopher.barnes@ucl.ac.uk

**Data Availability Statement:** Data and scripts required to reproduce the figures in this manuscript can be accessed from the following Zenodo repository: https://doi.org/10.5281/zenodo.5764686.

## Abstract

Predictability is a fundamental requirement in biological engineering. As we move to building coordinated multicellular systems, the potential for such systems to display chaotic behaviour becomes a concern. Therefore understanding which systems show chaos is an important design consideration. We developed a methodology to explore the potential for chaotic dynamics in small microbial communities governed by resource competition, intercellular communication and competitive bacteriocin interactions. Our model selection pipeline uses Approximate Bayesian Computation to first identify oscillatory behaviours as a route to finding chaotic behaviour. We have shown that we can expect to find chaotic states in relatively small synthetic microbial systems, understand the governing dynamics and provide insights into how to control such systems. This work is the first to query the existence of chaotic behaviour in synthetic microbial communities and has important ramifications for the fields of biotechnology, bioprocessing and synthetic biology.

## Author summary

In chaotic systems, infinitesimally small differences in the initial conditions will become amplified over time, making forecasting and prediction of behaviour impossible. Although we know that chaos can be observed in the complex networks of natural ecosystems, the field of biotechnology is interested in designing and building new microbial communities and the presence of chaotic behaviour is unexplored. In this paper, we present a statistical pipeline that can tell us how, when and why chaos arises in small microbial communities. We apply this approach to study a set of communities involving quorum sensing systems and amensal interactions through antimicrobial peptides. Out of 4182 interaction networks in these three strain communities, we identify the networks that have the highest propensity to produce chaos. We then explore the levers we can pull to bring these networks in and out of chaotic regimes. Our work is the first to look at chaos in synthetic microbial communities and indicates that chaos is an important design consideration.

**Funding:** B.D.K was funded through the BBSRC LIDo Doctoral Training Partnership, (Grant No 1758911). A.J.H.F and C.P.B received funding from the European Research Council (ERC) under the European Union's Horizon 2020 Research and Innovation Programme (Grant No. 770835). C.P.B. received salary funding from the Wellcome Trust (209409/Z/17/Z). The funders had no role in study design, data collection and analysis, decision to publish, or preparation of the manuscript.

## Introduction

Chaos can be defined as deterministic behaviour that displays aperiodic orbits and sensitivity to initial conditions [1]. Infinitesimally small differences in the initial conditions of a chaotic system will become amplified over time, making forecasting and prediction of behaviour impossible [2]. Despite being deterministic, chaotic systems possess an inherent uncertainty due to the fact that we can never describe the initial conditions of a system in sufficient detail. Building systems which behave in a predictable and repeatable manner is essential across fields invested in engineering biology and its applications. Evidence from studies of neural networks suggests the increasing probability of chaotic behaviour as the number of dimensions in the network grows [3–5]. Therefore we might expect opportunities for unpredictable behaviour to become more probable as we try and implement larger synthetic communities, or edit existing networks such as the human gut microbiome. Steps to date have not been taken to investigate the existence of chaos in small synthetic microbial networks. A long-term goal of engineering biology is to create truly scalable and robust synthetic microbial communities [6, 7]. Therefore understanding and evaluating the possibility of chaotic behaviour in a system becomes an important consideration.

Observations of chaotic behaviour in biological systems have been reported. A three species system containing one predator and two prey species has been demonstrated to produce chaotic behaviour, with dilution rate a key parameter in enabling aperiodic behaviours [8]. An eight year study of a planktonic food web measured chaotic behaviours, resulting in subpopulation abundance predictability being limited to 15–30 days, despite constant external conditions [9]. These experimental examples demonstrate that a low number of species are capable of producing chaotic behaviour and are therefore unpredictable.

In order to predict the possibility for chaotic behaviour in synthetic microbial communities, we need to develop models that capture interactions between different community species. Generalised Lotka-Volterra equations (gLV) have previously been used to model pair-wise interactions and infer inter-species relationships [10]. However, gLV models provide an incomplete description of interactions we expect to find in microbial communities. They are unable to capture the existence of chaos in three species networks [11]. Furthermore, gLV models have failed to predict community formation from pairwise interactions in microbial communities [12]. gLV models lack dynamics that occur with the accumulation and depletion of extracellular species, which can be important for predicting the true dynamics of a community [13]. Modified Lotka-Volterra equations produce chaotic behaviour in predator-prey systems by including time-delayed feedback [13, 14], or in one predator two prey systems, by adding dampening effects [15]. While these abstractions are suitable in some circumstances, by modelling the intermediates involved in competitive interactions we can include experimentally measurable mechanisms and parameters. In previous work, we have modelled quorum sensing (QS) to regulate bacteriocin expression and engineer inter-population interactions. These methods allowed us to tune experimental parameters of an engineered two strain system [16], and predict the most promising topologies for producing stability in two and three strain systems [17].

The existence of chaos in dynamical models and the distribution of chaotic parameter space can be identified using various optimisation techniques. The unscented Kalman filter has previously been used to investigate chaos in electrical circuits and biological systems, obtaining parameters yielding chaos [18]. Simulated annealing has been applied to finding chaotic parameters in four species standard Lotka Volterra models [19]. Evidence also suggests that perturbation of system parameters can be used to drive systems towards or away from

chaotic attractors [20]. The possibility of chaos in synthetic microbial communities, to our knowledge, has not been previously considered.

Standard competitive gLV models can produce chaotic behaviour in four species networks [11]. We use these previous findings to demonstrate and validate our methodology, before applying it to models that describe interpopulation interactions that are more specific to mechanisms found in synthetic microbial communities.

## Results

### Development of a novel statistical approach to identifying chaotic regions in a multidimensional parameter space

Approximate Bayesian Computation Sequential Monte Carlo (ABC SMC) is a method that can be used for model selection and parameter inference in dynamical systems [21] (Algorithm 1). This flexible algorithm can also be used to tackle the design question, namely what model topologies and parameters are able to give rise to some specified target qualitative behaviour [22]. ABC SMC requires a distance function, describing how far away a simulation is from the objective behaviour. When searching for chaotic beahviour, we use the maximal Lyapunov exponent ($\lambda_1$) to create a distance function. We calculate $\lambda_1$ by initialising two nearby orbits and measuring their divergence or convergence over the course of a simulation (see Methods and Algorithm 1). $\lambda_1 < 0$ corresponds to linear stability, $\lambda_1 = 0$ corresponds to periodic oscillations, and $\lambda_1 > 0$ corresponds to chaos. While these rules hold true for infinite time, our simulations run for a finite time, meaning these boundary rules can be noisy. To identify chaos, we therefore define a threshold above 0 where we can be sure simulations have chaotic behaviour.

First, we demonstrate the use of ABC SMC in resolving a chaotic parameter distribution in a competitive gLV system. Competitive gLV equations are commonly used in ecological population modelling, and have similarly been used to model microbial communities [23]. They describe generic negative interactions between species that could represent competition for nutrients or amensal interactions. Competitive gLV systems take the form

$$\frac{dN_i}{dt} = r_i N_i \left(1 - \sum_{j=1}^{n} \alpha_{ij} N_j\right)$$

where $N_i$ is the size of a species population, $r_i$ is the growth rate, $n$ is the number of species in the population and $\boldsymbol{\alpha}$ the interaction matrix, describes the amensal interactions between pairs of species in the system. To simulate the chemostat environment, we set the diagonal as a dilution rate, $D$, which is the same for all species. The diagonal of $\boldsymbol{\alpha}$ can also be thought of as defining the carrying capacity of each species.

$$\boldsymbol{\alpha} = \begin{bmatrix} D & \alpha_{12} & \alpha_{13} & \alpha_{14} \\ \alpha_{21} & D & \alpha_{23} & \alpha_{24} \\ \alpha_{31} & \alpha_{32} & D & \alpha_{34} \\ \alpha_{41} & \alpha_{42} & \alpha_{43} & D \end{bmatrix}$$

Vano et al. previously identified a chaotic attractor in this system using a brute-force parameter search [11]. Fig 1A shows the parameters identified, Fig 1B shows the resulting chaotic timeseries of the four species. We wanted to see if our ABC SMC methods could provide a posterior distribution for chaotic behaviour, capturing the findings of Vano et al. We

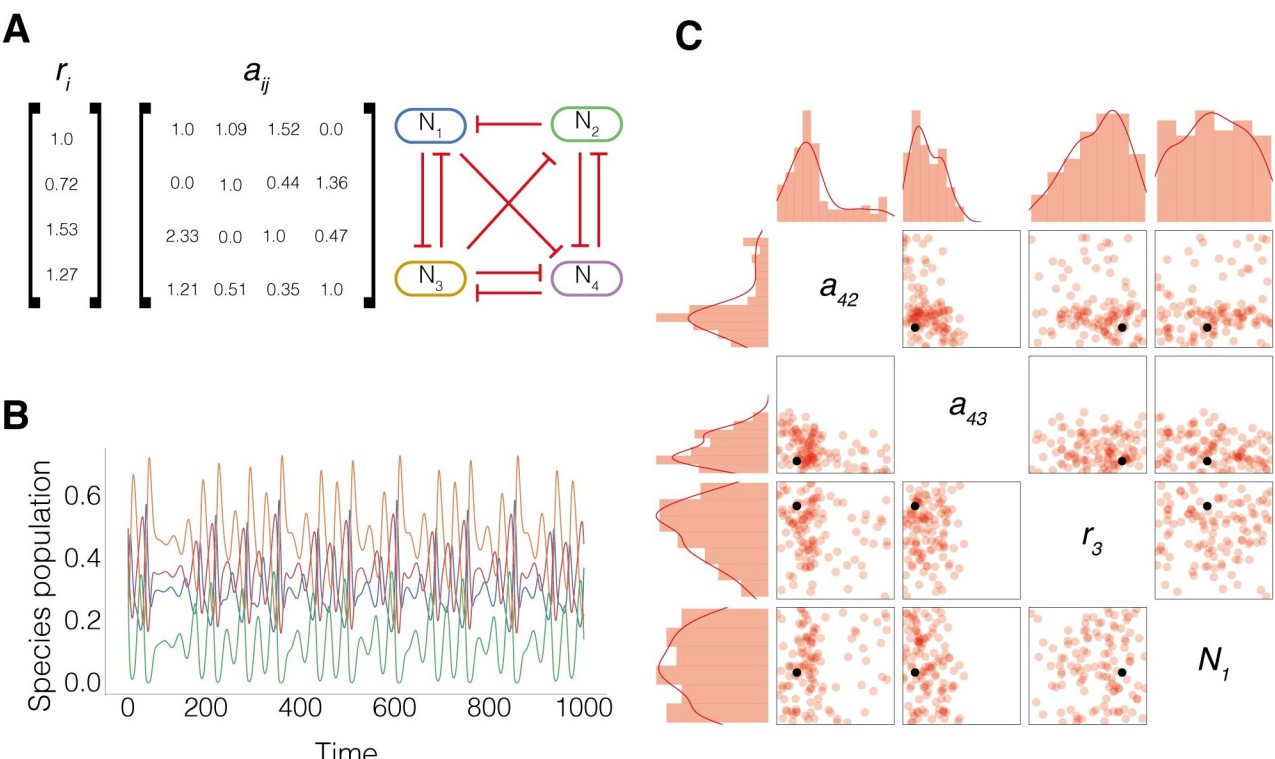

**Fig 1. Demonstration of chaotic attractor identified by Vano et al. in a four species competitive Lotka-Volterra model [11]. A** Shows the parameters used in the chaotic attractor and an illustration of the interaction topology. **B** Time series of the chaotic attractor. **C** Posterior parameter distribution for chaotic objective, identified using ABC SMC (red) and the individual chaotic particle identified by Vano et al. (black). Center grid shows 2D parameter distributions, left and top rows 1D parameter distribtuions.

identified a threshold of $\lambda > 0.015$ was sufficient for classifying chaotic behaviour and ran ABC SMC for this chaotic objective. We show the posterior of several parameters in Fig 1C. The black point corresponds to the parameters found by Vano et al, while the red scatter points correspond to chaotic behaviour we identified using ABC SMC. We can see that the interspecies interaction parameters, $a_{42}$ and $a_{43}$, are constrained, indicating their importance for producing chaotic behaviour, given the prior parameter distributions. Conversely, the initial population of $N_1$ is not constrained, indicating the chaotic behaviours are robust to changing initial conditions. Similarly, $r_3$, defining the growth rate of $N_3$ is not constrained. The full posterior parameter distribution is shown in S1 Fig.

Mechanisms of interaction in microbial communities such as crossfeeding and toxin interactions would be subjected to time delays, accumulation of intermediate species and dynamic genetic regulation, contributing to non-linearity of these systems. gLV equations simplify these mechanisms and as such, are unable to capture chaotic behaviour with three species. In the next sections we move to studying more biochemically realistic systems.

## Searching for chaos across synthetic microbial community models

In previous work we developed a model framework to describe QS regulated bacteriocin interactions in a three strain model space, and predicted topologies that form stable communities [17]. Here we use this same model space to investigate the existence of chaos in three strain synthetic microbial communities.

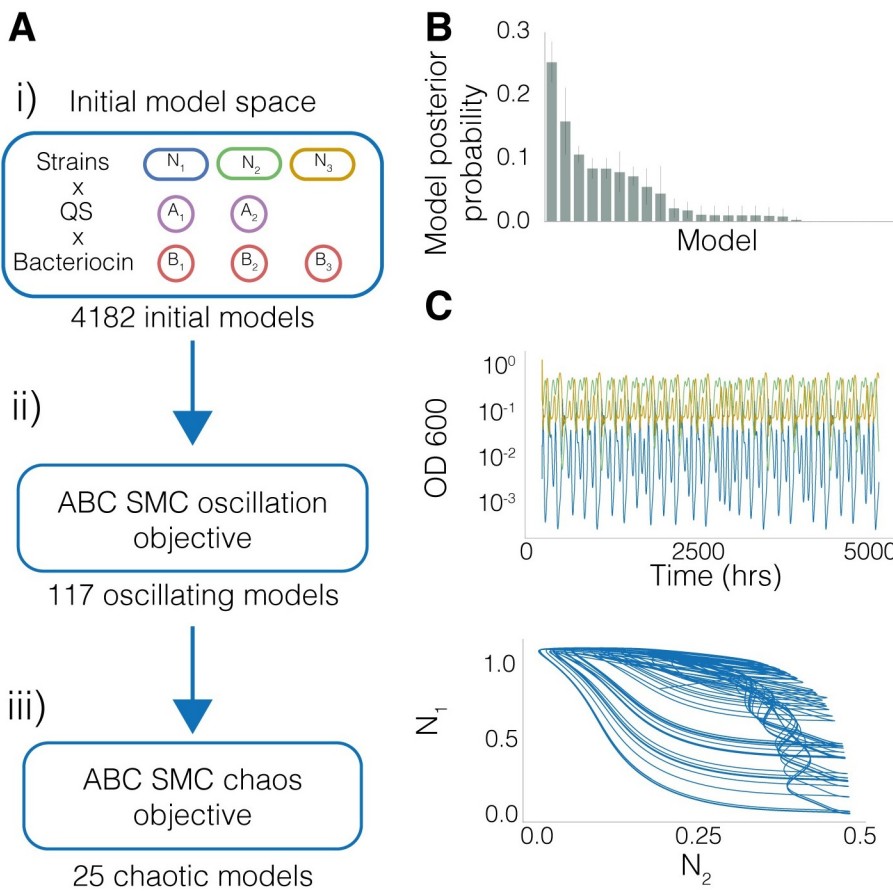

**Fig 2. Overview of the pipeline for identifying chaotic topologies. A(i)** The initial model space is built from different combinations of engineering options. $N_1, N_2, N_3$ are the three strains being engineered, and can optionally express QS molecules $A_1, A_2$ and bacteriocins $B_1, B_2, B_3$. 4182 models are generated forming our initial model space. **A (ii)** We then perform ABC SMC for an oscillatory objective, which yielded 117 models that were capable of producing oscillations. **A(iii)** These form the prior model space for the chaos objective, using a threshold of $\lambda_1 > 0.003$, we identify models capable of producing chaotic behaviour **B** The barchart shows the probability of models for the chaotic objective. The error bars represent the standard deviation. **C** An example time series representative of the chaos objective posterior distribution. Population densities as optical density (OD) show sustained, nonrepetitive oscillatory behaviour for the three species community.

Fig 2A shows the pipeline we developed to search for chaos in synthetic three strain systems. Three strains, $N_1, N_2, N_3$, optionally express bacteriocins, $B_1, B_2$, and $B_3$ under the control of optionally expressed QS molecules, $A_1$, and $A_2$. The QS molecules regulate expression of bacteriocins positively or negatively. Each strain can be optionally sensitive to a bacteriocin. The initial model space describes an enumeration of possible combinations of bacteriocin and QS systems, and forms the first uniform prior model space of 4182 models (Fig 2A(i)). Prior parameter distributions describe the range of characteristics for the different parts (Table 1). We expected the existence of chaos to be sparse in this three strain model space, and therefore computationally expensive to explore. Oscillations are a known route to chaos [1], therefore, in order to narrow down the search, we defined a novel set of three distances that are used to classify oscillatory behaviours. These were the period of the signal (defined through the Fourier transform), the number of expected peaks, and the amplitude of the signal (see Methods). We also define an extinction threshold of $10^{-5}$; if a strain population falls below this it is classified as extinct. Using these distances, we performed ABC SMC for an oscillations objective

**Table 1. Prior distributions for both two and three strain systems are sampled uniformly between the min and max values listed below.** Constant parameters have the same min and max value.

| Parameter / State variable | Description | Prior (min) | Prior (max) | Units | Citation |
|---|---|---|---|---|---|
| | Parameters | | | | |
| $C_N$ | OD to cell number scaling factor | $1e9$ | $1e9$ | None | N/A |
| $C_B$ | Microcin scaling factor | $1e{-}9$ | $1e{-}9$ | None | N/A |
| $C_A$ | QS scaling factor | $1e{-}9$ | $1e{-}9$ | None | N/A |
| $D$ | Dilution rate | 0.01 | 0.5 | $h^{-1}$ | N/A |
| $K_{A_y B_z}$ | Half maximal QS promoter activation/repression from $A_y$ to $B_z$ | $1e{-}9$ | $1e{-}6$ | $M$ | [43] |
| $K$ | Monod's half saturation constant | $3.9e{-}5$ | $3.9e{-}5$ | $M$ | [44] |
| $K_\omega$ | Half saturation killing constant | $1e{-}7$ | $1e{-}6$ | $M$ | [45, 46] |
| $S_0$ | Substrate concentration of input media (0.4% glucose) | 0.02 | 0.02 | $M$ | M9 media |
| $\gamma$ | *E. coli* substrate yield | $1e11$ | $1e11$ | cell $M^{-1}$ | [47] |
| $k_{A_y}$ | Production rate of AHL per cell | $1e{-}22$ | $1e{-}15$ | $M\,h^{-1}$ | [48] |
| $KB_{max}z$ | Maximal expression rate of microcin | $1e{-}22$ | $1e{-}15$ | $M\,h^{-1}$ | [49] |
| $\mu_{x_{max}}$ | Maximum growth rate | 0.4 | 3 | $h^{-1}$ | [50, 51] |
| $n_z$ | Hill coefficient AHL induced expression | 1 | 2 | $M$ | [43] |
| $n_\omega$ | Hill coefficient for killing | 1 | 2 | $M$ | [43] |
| $\omega_{max}$ | Maximum rate of bacteriocin killing | 0.5 | 2.0 | $M^{-1}\,h^{-1}$ | [45, 46, 52] |
| | Initial state variable | | | | |
| $N$ | OD of strain | 0.01 | 0.5 | OD | N/A |
| $S$ | 0.4% glucose concentration | 0.02 | 0.02 | M | N/A |
| $B$ | Microcin concentration | $1e{-}81$ | $1e{-}81$ | M $C_B$ | N/A |
| $A$ | QS concentration | $1e{-}10$ | $1e{-}10$ | M $C_A$ | N/A |

(Fig 2A(ii)). We identified 117 models capable of producing oscillations. These models become the new uniform prior model distribution for the next stage, where we perform ABC SMC for the previously described chaotic objective (Fig 2A(iii)). In this model framework we identified $\lambda_1 > 0.003$ as sufficient for classifying chaos.

The posterior probabilities of the models for the chaotic objective given the prior distributions used are shown in Fig 2B. Fig 2C shows a representative chaotic trajectory, demonstrating aperiodic non-repeating behaviour, satisfying the qualitative features of chaos.

## Properties of chaotic models

We next explored some of the properties of chaotic topologies identified using ABC SMC. Fig 3A shows the top performing models when subsetting for complexity, based on the number of parts expressed. $m_k$ refers to the $k$-th model from the initial model space. $m_{850}$ contains four expressed parts and possesses the highest posterior probability for chaotic behaviour. Systems containing fewer parts all had a posterior probability of zero. As complexity increases to five and six parts ($m_{3177}$ and $m_{2547}$), the posterior probability decreases. Our previous work demonstrated that system stability increased with the number of parts [17]. The peak in the posterior probability for chaos at four parts reflects a balance that includes enough mechanisms to enable co-existence, without the tighter network of negative interactions that are associated with linear stability [17, 24]. We highlight that these observations are limited to the small communities defined in our prior. These properties may not be reflective of larger communities, however, we hypothesise that a trade-off between stabilizing interactions that enable co-existence, and destabilising interactions to prevent linear stability, will remain important for producing chaotic population dynamics.

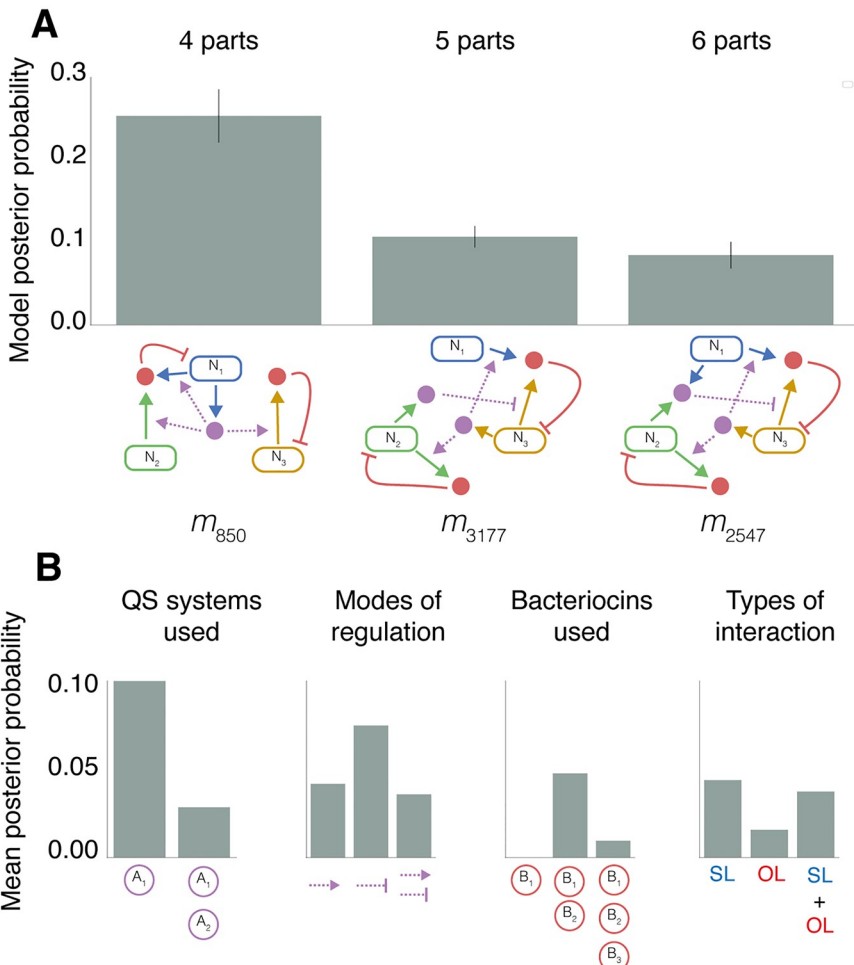

**Fig 3. Topologies and properties associated with chaotic behaviour.** **A** Shows the models with highest posterior probability when subsetted for number of parts expressed, in order of increasing complexity (4, 5 and 6 expressed parts). The bar chart shows the mean model posterior probability across three experiments, the error bars indicate the standard deviation. **B** Comparison between average posterior probabilities with different properties. In order from left to right, the barcharts compare: The number of QS systems used, the modes by which QS regulates bacteriocin expression (positive, negative or both), the number of bacteriocins used, and systems containing self-limiting (SL), other-limiting (OL) or SL and OL interactions.

Fig 3B provides summaries of how different parts contribute to chaotic behaviour in the three strain models. We can see that one QS system and positive regulation of bacteriocin is strongly favoured for producing chaos. This ensures all system bacteriocins are regulated in tandem. Expression rates are all dependent upon the same QS, resulting in stronger negative or positive correlations defined by the mode of regulation. Two bacteriocin systems also dominate the model posterior. Bacteriocin interactions can be categorised as either self-limiting (SL), whereby the strain is inhibited by the bacteriocin it produces, or other-limiting (OL) where a strain is inhibited by a bacteriocin produced by a different strain. Both SL only and a combination of SL and OL interactions are associated with producing chaotic behaviour. These observations are interesting in comparison to other work on ecological systems. Cooperative interactions were previously found to give rise to unstable systems, whereas competition was more indicative of stability [24]. The same effect might occur here in systems with

one QS, rather than two, as the system would be expected to have increased correlation. While chaotic behaviour may seem to be very different from linear stability, both behaviours share the necessity for coexistence. Our previous work showed that SL interactions were important for producing linear stability, while OL interactions more frequently associated with extinction events and non-linear stability [17]. This may explain why we see tendencies for topologies to share a mixture of stability associated SL interactions, and instability associated OL interactions. We also find models with three bacteriocins, and hence higher suppression of growth, have a low posterior probability for chaos, given the prior distributions used.

### Parameter importance for chaos in $m_{850}$

The model with the highest posterior probability for chaotic behaviour was $m_{850}$; the topology is shown in Fig 4A. It consists of a single QS system, produced by $N_1$, that positively regulates two bacteriocins. $B_1$ is produced by $N_1$ and $N_2$ but it inhibits the growth of $N_1$ only. $B_2$ is produced by $N_3$ and inhibits the growth of $N_3$ only. The system in total consists of four expressed parts. $m_{850}$ also ranked highly for the oscillatory objective, ranking 3rd out of the initial 4182

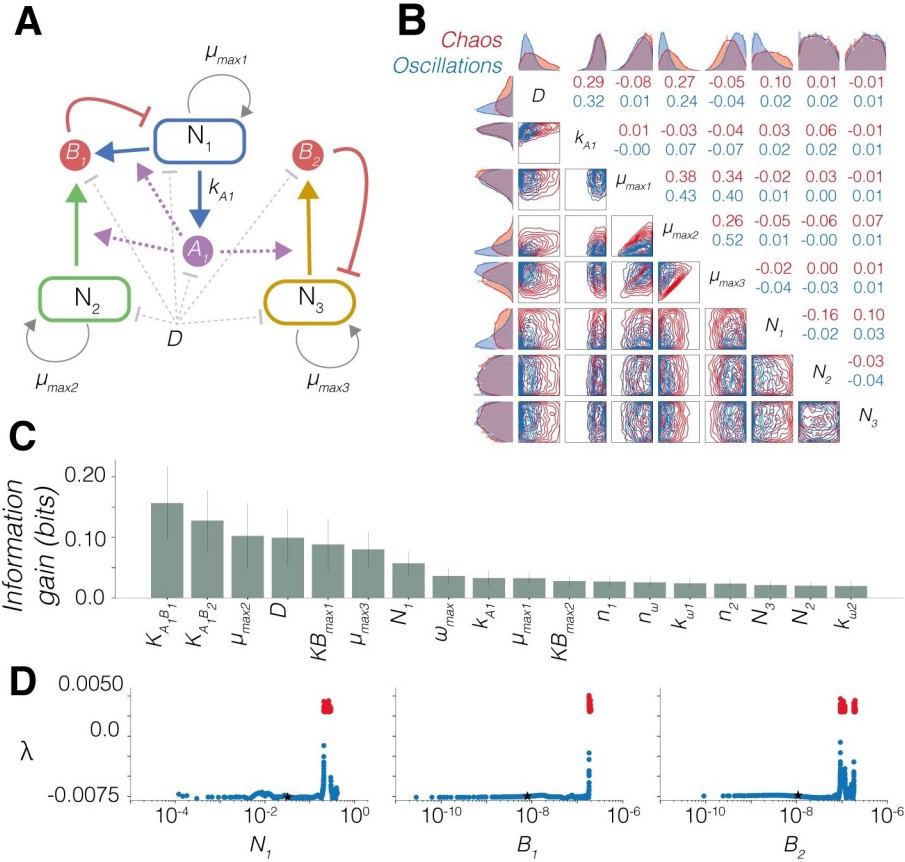

**Fig 4. Examining chaos in $m_{850}$. A** Topology of $m_{850}$ with key parameters labelled. $k_{A1}$ is the rate of QS molecule production, $KB_{max1}$ and $KB_{max2}$ are the maximal expression rates of bacteriocins $B_1$ and $B_2$ respectively. **B** Posterior parameter distributions of $m_{850}$ for chaos (red) and oscillatory (blue) objectives for key parameters in system design. The borders show 1D posterior distributions for each parameter and the lower-diagonal element the 2D posterior marginals, and the upper-diagonal shows the Pearson correlations. **C** Feature importance calculated using random forest regression. The information gain (bits) is calculated as an average of the reduction in entropy across all trees in the forest (2000 trees). The error bars indicate the standard deviation of the entropy for each feature across all trees. **D** Sensitivity analysis of a chaotic input vector with chaotic region in red. Black stars refer to the identified stable steady state. The fixed parameter values are shown in Table 2

**Table 2. Fixed parameters used in Figs 4D and 5.**

| Parameter/State variable | value |
|---|---|
| Parameters | |
| $C_N$ | $1e9$ |
| $C_B$ | $1e-9$ |
| $C_A$ | $1e-9$ |
| $D$ | $0.167$ |
| $K_{A_1 B_1}$ | $3.37e-9$ |
| $K_{A_1 B_1}$ | $4.26e-8$ |
| $K$ | $3.9e-5$ |
| $K_\omega$ | $1.6e-7$ |
| $S_0$ | $0.02$ |
| $\gamma$ | $1e11$ |
| $k_{A_y}$ | $3.5e-17$ |
| $KB_{max}1$ | $3.58e-17$ |
| $KB_{max}2$ | $8.89e-16$ |
| $\mu_{1_{max}}$ | $2.61$ |
| $\mu_{2_{max}}$ | $1.17$ |
| $\mu_{3_{max}}$ | $1.48$ |
| $n_1$ | $1.2$ |
| $n_2$ | $1.43$ |
| $n_\omega$ | $1.87$ |
| $\omega_{max}$ | $0.79$ |
| Initial state variable | |
| $N_1$ | $0.24$ |
| $N_2$ | $0.25$ |
| $N_3$ | $0.27$ |
| $S$ | $0.02$ |
| $B_1$ | $1e-71$ |
| $B_2$ | $1e-71$ |
| $A_1$ | $1e-10$ |

models. This presents an interesting problem whereby a model that has promising use as an oscillator also has a high potential to produce chaos, relative to other candidate models. Identifying the parameters and initial conditions important for differentiating between chaotic and oscillatory behaviour gives us insight into how to control this behaviour when constructing genetic circuits or selecting chemostat settings.

As a first step, we analyzed the model to quantify the possible steady states and basins of attraction. Our analysis gave analytical conditions for the existence and stability for complete extinction and for single strain survival (See Methods). For three-strain co-existence, we find the following necessary conditions:

$$\max\{D\frac{K+S_0}{S_0}, \mu_{1_{max}}\frac{D}{D+\omega_{max}}, \mu_{3_{max}}\frac{D}{D+\omega_{max}}\} < \mu_{2_{max}} < \min\{\mu_{1_{max}}, \mu_{3_{max}}\}$$

This shows that for three-strain co-existence, the maximal growth rate of $N_2$ has to lie between certain upper and lower bounds. In particular, it has to be smaller than the maximal growth rate of $N_1$ or $N_3$. We can see from the topology of $m_{850}$ (Fig 4A) that the growth of $N_2$ is not limited by any bacteriocin, therefore the only limitation on growth comes through

resource competition. If $N_2$ had a higher growth rate than $N_1$ or $N_3$, it would out compete these strains and cause an extinction event.

We then wanted to explore the most important parameters that separate oscillatory and chaotic behaviours in $m_{850}$ only. We refer to a set of parameters and initial conditions as an input vector. Using ABC SMC, we performed parameter inference on $m_{850}$ for the chaotic and oscillatory objectives, generating 3750 input vectors for each objective. We can use this dataset of labelled input vectors to understand the importance of individual parameters, initial conditions and nearby steady states.

Fig 4B shows multivariate parameter distributions for the oscillator and chaotic objectives for the experimentally accessible parameters. The dilution rate ($D$) is a directly controllable parameter of the chemostat. The production rate of $A_1$ ($kA_1$) can be tuned by using an inducible promoter to control expression of the AHL synthase species. Strain maximal growth rates ($\mu_{max1}$, $\mu_{max2}$, $\mu_{max3}$) can be controlled by using different base strains or through the combined use of auxotrophic strains and defined media. Finally, the initial population densities ($N_1$, $N_2$, $N_3$) can easily be set when inoculating the initial culture. Divergence between two parameter distributions indicates its importance in differentiating between the two objectives. We can see that the oscillatory objective distributions for $D$, $N_1$ and $\mu_{max2}$ are all constrained towards lower values relative to the prior. However, for all these distributions we can see that the chaotic and oscillatory regions overlap. This again implies that chaotic and oscillatory behaviour exist close to one another in parameter space, and highlights the multidimensional nature that determines the behaviour.

To further investigate the importance of parameters and initial conditions we trained a random forest classifier model using the input vectors as features [25]. We curated a label-balanced dataset of oscillating input vectors and chaotic input vectors. Using a train test ratio of 0.5, the trained classifier model was able to classify the test set with a $\sim$90% accuracy (Methods). Fig 4C shows the average information gain across all decision tree classifiers in the forest for all free parameters. This can be used as an indicator of feature importance in correctly classifying an input vector. $K_{A_1B_1}$ and $K_{A_1B_2}$ describe the concentration of $A_1$ required to produce half-maximal repression of bacteriocins $B_1$ and $B_2$ respectively. While the feature importance indicates these parameters are the most important, they are more difficult to tune compared with other parameters in this system. The error bars indicate the variability in the importance of a feature across all trees in the forest. Large error bars suggest single features are not essential for classification, and that redundancy exists between the features used [26].

From the set of chaotic input vectors, we used numerical methods to identify nearby steady states that could be reached by changing the initial state of the system only. Fig 4D shows the sensitivity analysis of a chaotic input vector. The black stars indicate stable steady states identified by numerical analysis. We perturbed the initial species values of either $N_1$, $B_1$ or $B_2$ individually. The plots show how changing these initial states yields different Lyapounv exponents, highlighting the chaotic region in red. The range of Lyapunov exponents shown in Fig 4D suggest that by changing the initial conditions only we are able to produce a range of different behaviours. Perturbing $N_2$, $N_3$ or $A_1$ did not produce chaotic behaviour. It is interesting that the initial state of $N_1$ as the $A_1$ producing strain appears to be more important whereas the initial concentration of $A_1$ itself is not.

## Exploring the parameters in the transition to chaos

Being able to move a system from a chaotic state to a fixed point could be important in a bioprocess control scenario so we explored this in more detail. Previous studies have frequently identified the bioreactor dilution rate as an important parameter for transitioning

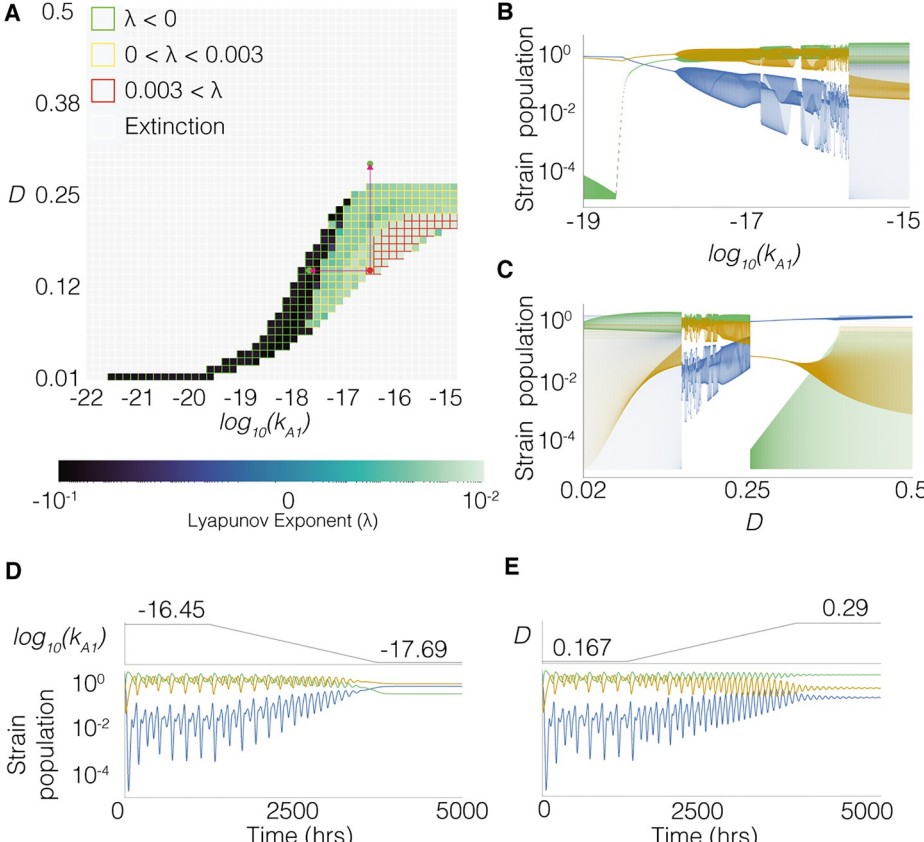

**Fig 5. Parameters $k_{A1}$ and $D$ can be tuned to control transitions between chaotic, oscillatory and stable states.** The fixed parameter values are shown in Table 2. **A** Map showing how different combinations of $k_{A1}$ and $D$ change population behaviour. The grid fill colour corresponds to the maximum Lyapunov exponent measured, the grid outlines indicate the approximate classification where green is stable, yellow is oscillatory, red is chaotic and white is extinction. **B** Bifurcation diagram showing the community states visited for different values of $k_{A1}$. **C** Bifurcation diagram showing the community states visited for different values of $D$. **D** Real-time ramp down tuning of $k_{A1}$, moving the system from a chaotic state to a stable steady state. **E** Real-time ramp up tuning of $D$, moving the system from a chaotic state to a stable steady state.

between different population dynamics [27–29]. The posterior parameter distribution shown in Fig 4B strongly indicated the dilution rate, $D$, to be important for defining chaotic behaviour. We previously identified the QS production rate, $k_{A1}$ and $D$ as important parameters for transitioning between co-existence and extinction states [16]. We hypothesised that the antagonistic effect of $k_{A1}$ to $D$ would make it a useful parameter for controlling population behaviour.

First, we took an input vector known to produce chaotic behaviour and randomly sampled new values for $k_{A1}$ and $D$ from the prior and calculated the Lyapunov exponent of the new input vector. Fig 5A shows the results where filled colour indicates the maximal Lyapunov exponent calculated at each grid reference. The grid outline indicates the classification, the red grid region of Fig 5A shows the chaotic region. Fig 5A illustrates that, changing $D$ and $k_{A1}$ affects the Lyapunov exponent. The bifurcation diagrams in Fig 5B and 5C for $k_{A1}$ and $D$ respectively, illustrate the antagonistic transitions in behaviour that occur when changing the two parameters. Fig 5A and 5B show transitions through one strain extinctions ($N_x < 10^{-5}$), stable steady state, oscillations and chaotic behaviour. Fig 5A and 5B both show that increasing $k_{A1}$ results in transitions from stable co-existence, through oscillations and then to chaos,

followed abruptly by an extinction event. Fig 5A and 5C and c both show that a lower dilution rate is associated with chaos; increasing the dilution rate reduces instability to produce oscillations, which abruptly transitions to a stable extinction state.

In a bioreactor control scenario it is interesting to understand if a community could be switched between states in real time. Fig 5D and 5E show how this is possible by modifying $k_{A1}$ and $D$ respectively. The red arrows on Fig 5A indicate the position of the single start point and two end points in these real-time transitions. It is important to note that when ramping up the dilution rate in real-time, we reach stable steady state in a region that would not be obtainable with a fixed dilution rate.

## Discussion

We have developed a novel methodology to explore parameter regions that give rise to chaotic dynamics. We have applied it to the exploration of chaotic dynamics in synthetic microbial communities and found a high prevalence of such dynamics in these systems. This work is the first to query the existence of chaotic behaviour in synthetic microbial communities. We show that we can expect to find chaotic states in relatively small synthetic microbial systems, which has important ramifications for the field.

By first running ABC SMC for the oscillatory objective we were able to drastically reduce the model space for the search for chaos. However, the timecourse simulation and parameter sampling makes this pipeline computationally costly. In the future we can consider using eigenvalue stability methods to reject particles without simulation, improving the efficiency of our approach and therefore improving the number of samples available for posterior estimation [30, 31].

We expect it will become increasingly important to consider the location of chaotic attractors in parameter space as the microbial communities we build or interact with become more complex. These methods can easily be applied to parametrise different models. It would be interesting to compare the existence of chaotic attractors in systems that use toxin-antitoxin systems [32], combinations of cooperative and competitive interactions [33], or mutualistic only interactions [34]. Genome scale metabolic models contain a large number of linear reactions [35]; they can be combined to describe microbial communities and used to model industrial bioprocesses [36, 37]. Given the high dimensional nature of metabolic networks, it would be interesting to investigate whether these models yield chaotic behaviour in small community networks.

## Conclusion

To conclude, we have developed methods for identifying chaotic parameter regions using ABC SMC. We have demonstrated the application of this method to resolve a previously identified chaotic attractor in a gLV model, and identified models susceptible to chaos in three strain synthetic microbial communities. Although chaotic attractors are generally thought to be sparse in low dimensional systems, we have shown their existence in realistic synthetic microbial systems. They may also exist in close proximity to stable steady state regions. This work demonstrates that deterministic chaos will be an important factor in microbial community design and should be studied in much more detail.

## Materials and methods

### Three strain synthetic communities model space definition

Models are generated from a set of parts, that are expressed by different strains in the system. We represent an expression configuration through a set of options. We define the options for expression of $A$ in each strain, where the options are: not expressed, expression of $A_1$, or

expression of $A_2$ (0, 1 or 2 respectively). We define the options for expression of bacteriocin as: no expression, expression of $B_1$, expression of $B_2$ or expression of $B_3$ (0, 1, 2 and 3 respectively). Lastly we define the mode of regulation, $R$, for each bacteriocin, which can be either induced or repressed (0 and 1). This is redundant if a bacteriocin is not expressed.

$$A = \{0, 1, 2\}$$
$$B = \{0, 1, 2, 3\}$$
$$R = \{0, 1\}$$

This enables us to build possible part combinations that can be expressed by a population. Let $P_c$ be a family of sets, where each set is a unique combination of parts.

$$P_C = A \times B \times R$$

Each strain in a system can be sensitive to up to one bacteriocin. Let $I$ represent the options for strain sensitivity. The options are: insensitive, sensitive to $B_1$, sensitive to $B_2$ or sensitive to $B_3$ (0, 1, 2 and 3 respectively).

$$I = \{0, 1, 2, 3\}$$

Each strain is defined by its sensitivities, and expression of parts. Let $P_E$ be all unique engineered strains:

$$P_E = I \times P_C$$

Which can be combined to form a model, yielding unique combinations:

$$P_M = P_E \times P_E \times P_E$$

Finally, we use a series of rules to remove redundant models. A system is removed if:

1. Two or more strains are identical, concerning bacteriocin sensitivity and combination of expressed parts.

2. The QS regulating a bacteriocin is not present in the system.

3. A strain is sensitive to a bacteriocin that does not exist in the system.

4. A bacteriocin exists that no strain is sensitive to.

   This cleanup yields the options which are used to generate ODE equations for a system.

## System equations

State variables in each system are rescaled to improve speed of obtaining numerical approximations. $N_X$ describes the concentration of a strain, $B_z$ describes the concentration of a bacteriocin and $A_y$ describes the concentration of a quorum molecule. $C_N$, $C_B$ and $C_A$ are scaling factors:

$$N'_x = N_x C_N$$
$$B'_z = B_z C_B$$
$$A'_y = A_y C_A$$

Each model is represented as sets where $\mathbb{N}$ defines the number of strains, $\mathbb{B}$ defines the set of bacteriocins and $\mathbb{A}$ defines the set of QS systems. The following differential equations are

used to represent each model.

$$\frac{dN_x}{dt} = N_x \mu_x(S) - N_x \sum_{z=1}^{\mathbb{B}} \omega(B'_z) - N_x D$$

$$\frac{dS}{dt} = D(S_0 - S) - \sum_{x=1}^{\mathbb{N}} \frac{\mu_x N'_x}{\gamma}$$

$$\frac{dB_z}{dt} = \sum_{x=1}^{\mathbb{N}} \frac{(k_{B_{x,z}} N'_x)}{C_B} - D B_z$$

$$\frac{dA_y}{dt} = \sum_{x=1}^{\mathbb{N}} \frac{k_{A_{x,y}} N'_x}{C_A} - D A_y$$

Growth is modelled by Monod's equation for growth limiting nutrient, $S$. $\mu_{x_{max}}$ defines the maximal growth rate of the strain and $K_X$ defines the concentration of substrate required for half-maximal growth.

$$\mu_x(S) = \frac{\mu_{x_{max}} S}{K_X + S}$$

Killing by bacteriocin is defined by $\omega(B'_z)$, where $\omega_{max}$ defines the maximal killing rate which is set to 0 if the strain is insensitive. $K_\omega$ defines the concentration at which half-maximal killing occurs.

$$\omega(B'_z) = \omega_{max} \frac{B'^{n_\omega}_z}{K_\omega^{n_\omega} + B'^{n_\omega}_z}$$

Induction or repression of bacteriocin expression by QS, is defined by $k_B(z, y)$, where $z$ defines the bacteriocin being expressed and $y$ defines the quorum molecule regulating its expression. $KB_{max} z$ is the maximal expression rate of the bacteriocin and $K_{B_z}$ is the concentration of quorum molecule at which bacteriocin is half-maximal. $n_z$ defines the cooperativity of the AHL binding.

$$k_B(z, y) = KB_{max} z \frac{A'^{n_z}_y}{K_{B_z}^{n_z} + A'^{n_z}_y}$$

$$k_B(z, y) = KB_{max} z \frac{K_{B_z}^{n_z}}{K_{B_z}^{n_z} + A'^{n_z}_y}$$

## Software packages and simulation settings

The ABC SMC model selection algorithm was written in python using Numpy [38], Pandas and Scipy [39]. ODE simulations were conducted in C++ with a Rosenbrock 4 stepper from the Boost library [40]. All simulations use an absolute error tolerance of $1e{-}9$, and relative error tolerance of $1e{-}4$. Simulations were conducted for 5000hrs, and were stopped early if the population of any strain fell below $1e{-}5$ (extinction event). Simulations with an extinction event have distances set to maximum in order to prevent excessive time spent simulating collapsed populations.

## Approximate Bayesian computation with sequential monte-carlo (ABC SMC)

Particles are first sampled from the prior distribution and simulated. A set of distances, $d$, are calculated from the simulation. If all distances are less than the intermediate threshold, $\epsilon_t$, the particle is accepted ($d < \epsilon_t$). Accepted particles are weighted using importance sampling. The next population is sampled from the previous, and a new threshold is generated that is closer to the final threshold, $\epsilon_F$. This process is repeated until we reach a distance threshold of $\epsilon_F$. ABC SMC is highly parallel, allowing us to take advantage of high performance computing resources [21].

**Updating $\epsilon_t$.** At the end of each population of ABC SMC, the distance threshold $\epsilon_t$ is updated to approach the final population, $\epsilon_F$. The quantile parameter, $\alpha$, is defined. The distances of the population are sorted in ascending order and the distance at quantile $\alpha$ is used as the threshold for the next population. If $\epsilon_t < \epsilon_F$, we set $\epsilon_t = \epsilon_F$, marking the next population as the final population.

**Algorithm 1:** Algorithm for model selection with ABC SMC

```
1: Initialisation
   Set population indicator, t = 0
   Set ϵt
   Set final epsilon, ϵF
   Set population size, N
   Set population particle count, i
   Set distance threshold quantile, α
2: Sample particle, consisting of a model (m) and parameters (θ):
   If t = 0, sample θ**(m*) from prior distribution, π(m, θ)
   If t > 0, sample θ*(m*) from previous distribution {θ(m*)ⁱₜ₋₁} with
   weights w(m*)ₜ₋₁
3: Perturb particle
   If t > 0, perturb particle using perturbation kernel Kₜ, yielding
   perturbed particle θ**(m*)
4: Simulate particle
   x* ~ f(x|θ*, m*)
5: Calculate distance from objective
   d = ρ(x*, x₀)
6: Accept or reject particle
   If d > ϵt, reject particle and go to 2
   If d < ϵt, accept particle, add θ**, m* to population {θ(m*)ⁱₜ}
7: Set accepted particle weight
   Particle weight, w, is set to 1 for the initial population. For sub-
   sequent populations, the weight of a particle is equal to the proba-
   bility of observing the particle given the prior, divided by the
   probability of observing the particle given the previous population.
```

$$w_t^j = \frac{\pi(\theta^{**})}{\sum_{j=1}^{N} w_{t-1}^j K_t(\theta_{t-1}^i | \theta^{**})}$$

```
   i = i + 1
   If i < N go ot 2
8: Population full
   Normalise population particle weights
   If ϵt == ϵF, return {θ(m*)ⁱₜ} and wₜ, the approximation of the posterior
   distribution
   Prepare next population
   Set i = 0
   Set t = t + 1
   Update the distance threshold as a function of the distances in the
   population, d and the threshold quantile, α, ϵt = fₑ(α, d)
   go to 2
```

## Oscillatory population dynamic objective

We define the oscillatory population dynamic using three summary statistics for each strain. First, we use Fourier transform of the population signal to find the maximum frequency, $f$, and convert this to the period, T.

$$T = 1/f$$

We set a minimum period of $t/2$ where $t$ is the simulation time, giving us $d_{o_1}$. $d_{o_1}$. Any simulations in which $T < t/2$, $d_{o_1}$ is set to 0, this distance ensures that all we have frequencies of oscillations that are on a scale relevant to the time period being measured. It was found that using the signal frequency alone resulted in acceptance of many simulations with very small oscillations, or simulations that rapidly dampen. We therefore generated two additional distances that account for oscillation amplitudes to select for sustained oscillations only. We can define the number of expected peaks in the simulation, $p$.

$$p = \frac{t}{T}$$

Peaks in the trajectory are identified by changes from a positive gradient to a negative gradient, and troughs via changes from negative gradient to positive gradient. The peak-to-peak amplitudes are calculated by differences between consecutive peaks and troughs. $A_K$ is the number of amplitudes above the threshold, $K = 0.05$. $d_{o_2}$ is the difference between the number of expected oscillations in the simulation, and the count of above threshold oscillations. Because incomplete oscillations at the time the simulation ends can impact the distance measurement, we set a lenient final distance threshold for $d_{o_2}$. $d_{o_3}$ compares the final amplitude $A_F$ in the simulation to the threshold. We set $d_{o_3} = 0$ if $d_{o_3} > K$.

$$
\begin{aligned}
d_{o_1} &= |T - t/2| \\
d_{o_2} &= |A_K - p| \\
d_{o_3} &= |A_F - K| \\
\boldsymbol{d} &= (d_{o_1}, d_{o_2}, d_{o_3}) \\
\boldsymbol{\epsilon}_F &= (2.0, 2.5, 20.0)
\end{aligned}
$$

## Maximal Lyapunov exponent calculation

Lyapunov exponents can be used to measure chaotic behaviour; they describe the average exponential rate of divergence between two near trajectories of a dynamical system. The maximal Lyapunov exponent, $\lambda_1$, can be used as determinant of chaotic behaviour. Using a method described by Sprott et al. [41], we evolve two nearby orbits and measure their average rate of separation. This directly investigates whether small changes to an initial state will produce a disproportionate separation. By periodically readjusting the distance of divergence after each time step we measure separation across a period of time, preventing a single event dominating subsequent states (Fig 6). The method is described in full by Algorithm 2. For all simulations we generate nearby orbits by perturbing one of the strain initial strain densities by $\triangle_0 = 10^{-10}$. All simulations use a transient time equivalent to the first 10% of the time series.

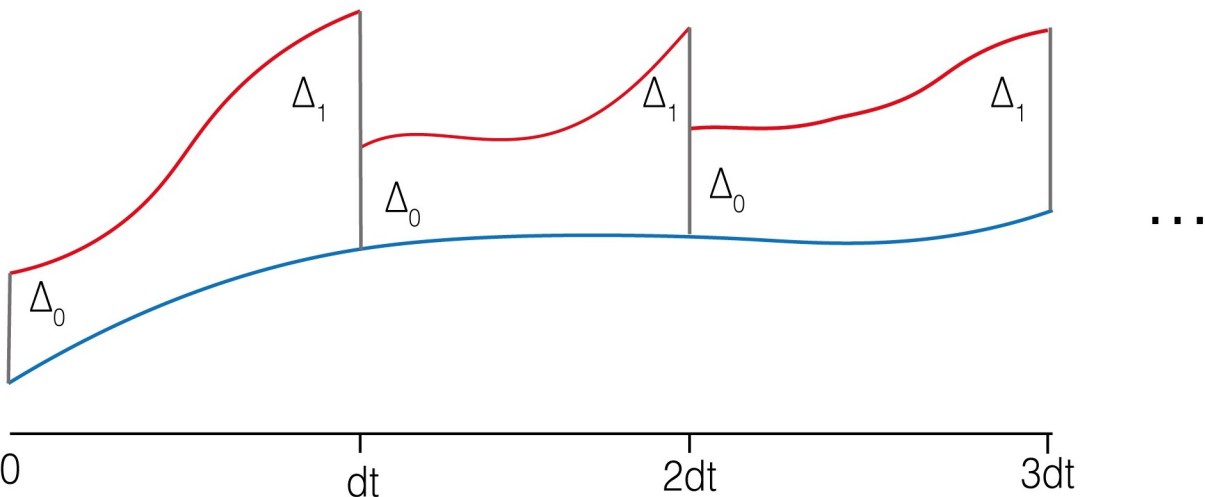

**Fig 6. Illustration of dual-orbit algorithm used to calculate the $\lambda_1$.** Two orbits with an initial state separation of $\triangle_0$ are followed. After each time step measure the separation, $\triangle_1$, is measured. The perturbed orbit (red) is readjusted to prevent excess separation. The average rate of separation between the two orbits corresponds with the $\lambda_1$.

## Chaos population dynamic objective

$d_{C_1}$ is the only distance for the chaotic objective. If $d_{C_1} < 0$, the particle is rejected. The final distance threshold, $\epsilon_C$, is equivalent to all $\lambda_1 > 0.003$.

$$d_{C_1} = 1/(1 + \lambda_1)$$
$$\boldsymbol{d} = (d_{C_1})$$
$$\boldsymbol{\epsilon}_C = \{0.997\}$$

For each sampled particle a prescreening process was performed to minimise time spent conducting the more computationally time consuming dual-orbit method. Simulations in which a strain fell below $10^{-5}$ were rejected. The number of oscillations with an amplitude greater than 0.05 was counted for each strain signal. If any strain showed less than 2 oscillations the particle was rejected. ABC SMC was conducted with population sizes of 10, repeated 255 times yielding a combined final population of 2550 particles.

## Random forest classifier model

Using the sci-kit learn (sklearn) python package [42], a random forest classifier was trained using 2000 estimators. The data used consisted of 3750 oscillatory input vectors, and 3750 chaotic input vectors. Training and test datasets were generated with a ratio of 0.5 by random sampling. Fig 7 shows the performance of the classifier model on the test data.

**Algorithm 2:** Description of dual-orbit method, demonstrated with two-dimensional system

```
1 Set S = 0.0
2 Set parameters and initial state θ_i = (x_i, y_i) for orbit, f(θ_i)
3 Simulate f(θ_i) for transient time, t_t, yielding state, θ_a0
4 Set initial state of nearby orbit, f(θ_b0), where, θ_b0 = θ_a0 + △_0
5 Set t = 0
6 Advance f(θ_a0) and f(θ_b0) by one step, dt, yielding states θ_a1 and θ_b1
respectively
7 Set t = t + dt
8 Calculate separation between the state variables of the two orbits,
△_1 = [(x_a1 − x_b1)² + (y_a1 − y_b1)²]^(1/2)
```

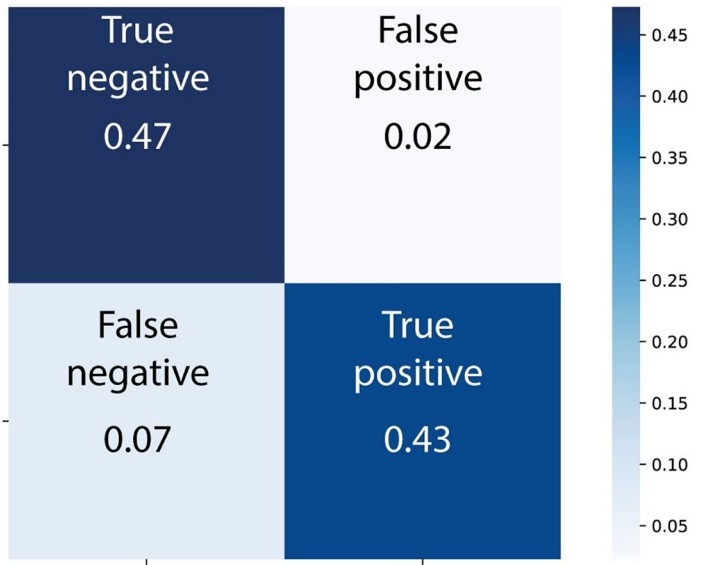

**Fig 7. Confusion matrix showing accuracy of random forest classifier on test data.**

```
9  S = S + log₂(|△₁/△₀|)
10 Readjust θ_{b₁} to align directionally with θ_{a₁}, x_{b₀} = x_{a₁} + △₀(x_{b₁} − x_{a₁})/△₁
and y_{b₀} = y_{a₁} + △₀(y_{b₁} − y_{a₁})/△₁
11 Set x_{a₀} = x_{a₁} and x_{b₀} = x_{b₁}
12 Repeat lines 6 to 11 for n iterations
13 Calculate maximal Lyapunov exponent as an average of the separation
values, λ₁ = S/n
```

## Analysis of $m_{850}$

$m_{850}$ is described by the following equations

$$\frac{dN_1}{dt} = N_1 \frac{\mu_{1_{max}} S}{K + S} - \omega_{max} \frac{N_1 B_1'^{n_\omega}}{K_\omega^{n_\omega} + B_1'^{n_\omega}} - N_1 D \tag{1a}$$

$$\frac{dN_2}{dt} = N_2 \frac{\mu_{2_{max}} S}{K + S} - N_2 D \tag{1b}$$

$$\frac{dN_3}{dt} = N_3 \frac{\mu_{3_{max}} S}{K + S} - \omega_{max} \frac{N_3 B_2'^{n_\omega}}{K_\omega^{n_\omega} + B_2'^{n_\omega}} - N_3 D \tag{1c}$$

$$\frac{dS}{dt} = D(S_0 - S) - \frac{\mu_1 N_1'}{\gamma} - \frac{\mu_2 N_2'}{\gamma} - \frac{\mu_3 N_3'}{\gamma} \tag{1d}$$

$$\frac{dB_1}{dt} = \frac{k_{B_{1,1}} N_1'}{C_B} - DB_1 \tag{1e}$$

$$\frac{dB_2}{dt} = \frac{k_{B_{2,1}} N_3'}{C_B} - DB_2 \tag{1f}$$

$$\frac{dA_1}{dt} = \frac{k_{A_{1,1}} N_1'}{C_A} - DA_1 \tag{1g}$$

$$N'_x = N_x C_N$$
$$B'_z = B_z C_B$$
$$A'_y = A_y C_A$$
$$k_{B_{z,y}} = KB_{max} z \frac{A'^{n_z}_y}{K^{n_z}_{B_z} + A'^{n_z}_y}$$

By setting the left hand side of (1) to 0 we find a number of steady states $\mathbf{P} = (N_1, N_2, N_3, S, B_1, B_2, A_1)$.

**The trivial steady state.** $\mathbf{P_0} = (0, 0, 0, S_0, 0, 0, 0)$. The Jacobian of the linearisation has eigenvalues

$$-D, \quad -\left(D - \mu_{1_{max}} \frac{S_0}{S_0 + K}\right), \quad -\left(D - \mu_{2_{max}} \frac{S_0}{S_0 + K}\right), \quad -\left(D - \mu_{3_{max}} \frac{S_0}{S_0 + K}\right).$$

Consequently the trivial steady state always exists and is linearly stable for

$$D > \frac{S_0}{S_0 + K} \max\{\mu_{1_{max}}, \mu_{2_{max}}, \mu_{3_{max}}\}.$$

This shows that if the dilution rate is high enough, no strain can survive.

**One strain only steady states.** There are three steady states where only one strain survives, $\mathbf{P_1}, \mathbf{P_2}, \mathbf{P_3}$. While $\mathbf{P_2}$, and $\mathbf{P_3}$ can be calculated explicitly, $\mathbf{P_1}$ is given implicitly (see below).

We start with $\mathbf{P_2}$:

$$\mathbf{P_2} = (0, N_2, 0, S, 0, 0, 0), \quad \text{where}$$
$$N_2 = \frac{\gamma}{C_N} \frac{S_0 \mu_{2_{max}} - D(S_0 + K)}{\mu_{2_{max}} - D}, \quad S = \frac{DK}{\mu_{2_{max}} - D}.$$

We see that $\mathbf{P_2}$ exists provided

$$D < \frac{\mu_{2_{max}} S_0}{S_0 + K}.$$

The linearisation at $\mathbf{P_2}$ has eigenvalues

$$-D, \quad -D\left(1 - \frac{\mu_{1_{max}}}{\mu_{2_{max}}}\right),$$
$$-D\left(1 - \frac{\mu_{3_{max}}}{\mu_{2_{max}}}\right),$$
$$-\frac{1}{K\mu_{2_{max}}}\left(\mu_{2_{max}} - D\right)\left(\mu_{2_{max}} S_0 - D(S_0 + K)\right).$$

This shows that $\mathbf{P_2}$ exists and is linearly stable if

$$D < \frac{\mu_{2_{max}} S_0}{S_0 + K}, \quad \text{and} \quad \mu_{2_{max}} > \max\{\mu_{1_{max}}, \mu_{3_{max}}\}.$$

The situation for $\mathbf{P_3}$ is very similar:

$$\mathbf{P_3} = (0, 0, N_3, S, 0, 0, 0), \quad \text{where}$$

$$N_3 = \frac{\gamma}{C_N} \frac{S_0 \mu_{3_{\max}} - D(S_0 + K)}{\mu_{3_{\max}} - D}, \quad S = \frac{DK}{\mu_{3_{\max}} - D}.$$

We see that $\mathbf{P_3}$ exists provided

$$D < \frac{\mu_{3_{\max}} S_0}{S_0 + K}.$$

The linearisation at $\mathbf{P_3}$ has eigenvalues

$$-D,$$

$$-D\left(1 - \frac{\mu_{2_{\max}}}{\mu_{3_{\max}}}\right),$$

$$-D\left(1 - \frac{\mu_{1_{\max}}}{\mu_{3_{\max}}}\right),$$

$$-\frac{1}{K\mu_{3_{\max}}}(\mu_{3_{\max}} - D)(\mu_{3_{\max}} S_0 - D(S_0 + K)).$$

This shows that $\mathbf{P_3}$ exists and is linearly stable if

$$D < \frac{\mu_{3_{\max}} S_0}{S_0 + K}, \quad \text{and} \quad \mu_{3_{\max}} > \max\{\mu_{1_{\max}}, \mu_{2_{\max}}\}.$$

The steady state $\mathbf{P_1} = (N_1, 0, 0, S, B_1, 0, A_1)$ is more complicated and can not be expressed explicitly. Instead it is given as follows: Assume there exists a solution $S$ to the following equation

$$\mu_{1_{\max}} \frac{S}{K + S} - D = \omega_{max} \frac{(B_1(S)C_B)^{n_\omega}}{K_{\omega_1}^{n_\omega} + (B_1(S)C_B)^{n_\omega}}, \tag{2}$$

where

$$B_1(S) = \frac{K_{Bmax_1}}{k_{A_1}} \frac{(A_1(S)C_B)^{n_1+1}}{K_{AB_1}^{n_1} + (A_1(S)C_B)^{n_1}}, \quad \text{and} \quad A_1(S) = \frac{k_{A_1}\gamma}{C_B \mu_{1_{\max}}} \frac{(S_0 - S)(K + S)}{S}.$$

If such a solution $S$ exists then $B_1 = B_1(S)$, $A_1 = A_1(S)$ and $N_1 = \frac{DC_B}{k_{A_1}C_N} A_1$.

**Lemma 1** *There exists a unique steady state $\mathbf{P_1}$ if and only if*

$$D < \mu_{1_{\max}} \frac{S_0}{K + S_0}.$$

**Proof:** We need the solution to (2) to fulfil $S < S_0$ in order for $A_1$ to be positive. We interpret the right and left-hand-sides of (2) as a function of $S$, denoting them by $R(S)$ and $L(S)$ respectively. It is easy to see that $A_1(S)$ is a decreasing function of $S$, $B_1(S)$ increases as a function of $A_1$ and $R(S)$ is an increasing function of $B_1$. Consequently the $R(S)$ is a decreasing function of $S$. We also see that $R(0) = \omega_{max} > 0$ and $R(S_0) = 0$. Further $L(0) = -D$ and $L(S_0) = \mu_{1_{\max}} \frac{S_0}{K+S_0} - D$ and $L(S)$ increases as a function of $S$. This proves the statement.

To summarise, the single-strain survival steady state requires the corresponding maximal growth rate to be large compared to other parameters.

**The three-strain co-existence steady state.** $P_{123} = (N_1, N_2, N_3, S, B_1, B_2, A_1)$.
From the equation for $N_2$ we obtain that

$$S = \frac{DK}{\mu_{2_{max}} - D}$$

$$\mu_{2_{max}} < \min\{\mu_{1_{max}}, \mu_{3_{max}}\},$$

$$D < \min\{\mu_{2_{max}} \frac{S_0}{K + S_0}, \omega_{max} \frac{\mu_{2_{max}}}{\mu_{1_{max}} - \mu_{2_{max}}}, \omega_{max} \frac{\mu_{2_{max}}}{\mu_{3_{max}} - \mu_{2_{max}}}\}$$

*Stability.* Solved numerically using MATLAB. For each of the 3750 chaotic input vectors we used numerical root finding to calculate $P_{123}$, and determined its stability by numerically determining the eigenvalues of the Jacobian. We found $P_{123}$ existed for all 3750 input vectors and was stable for 7.8% of them.

## Supporting information

**S1 Fig. Posterior distribution of chaotic objective for gLV model.** Posterior distribution of of all parameters.
(TIF)

## Author Contributions

**Conceptualization:** Behzad D. Karkaria, Alex J. H. Fedorec, Chris P. Barnes.

**Formal analysis:** Behzad D. Karkaria.

**Methodology:** Behzad D. Karkaria, Angelika Manhart, Alex J. H. Fedorec.

**Project administration:** Chris P. Barnes.

**Resources:** Chris P. Barnes.

**Software:** Behzad D. Karkaria.

**Supervision:** Chris P. Barnes.

**Visualization:** Behzad D. Karkaria.

**Writing – original draft:** Behzad D. Karkaria, Angelika Manhart.

**Writing – review & editing:** Behzad D. Karkaria, Angelika Manhart, Alex J. H. Fedorec, Chris P. Barnes.

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
