## [Decision Letter · Decision Letter 0]

8 Feb 2022

Dear Dr Barnes,

Thank you very much for submitting your manuscript "Chaos in small microbial communities" for consideration at PLOS Computational Biology.

As with all papers reviewed by the journal, your manuscript was reviewed by members of the editorial board and by several independent reviewers. In light of the reviews (below this email), we would like to invite the resubmission of a significantly-revised version that takes into account the reviewers' comments.

We cannot make any decision about publication until we have seen the revised manuscript and your response to the reviewers' comments. Your revised manuscript is also likely to be sent to reviewers for further evaluation.

Sincerely,

Christopher Rao

Associate Editor

PLOS Computational Biology

Jason Papin

Editor-in-Chief

PLOS Computational Biology

Reviewer's Responses to Questions

**Comments to the Authors:**

Reviewer #1: Here Karkaria and colleagues present a new method for exploring chaos within a in silico microbial communities. They apply this method to one particular family of in silico models (composed of 3 species, 2 QS systems, 3 bacteriocins) to identify a small subset of models in which chaos is most likely. They then explore in more depth how chaos depends upon parameter sets for one specific model, m850, identified by their initial screening process.

Overall, this paper reads as a thorough explanation of chaos in one particular in silico model. It’s well written, the figures are helpful, and the models were largely well explained - however, my impression is that the focus seems far too narrow to be of interest to the broad plos comp bio readership.

Whether a paper is of sufficient interest for the journal is up to the editor. However, I will try and outline the sort of thing I would expect to see to make this paper more broadly applicable, depending on what exactly the authors were aiming to achieve. For example,

a. Was the goal to present a new method for identifying the likelihood of chaos within microbial communities, ABC SMC? If so, I’d have expected to see some proof that it works – ie are there any empirical data out there the model can be compared against? Or, at the very least, I would expect to see the method applied to a far broader set of models (more species etc) to explore how scalable it is.

b. Alternatively, was the goal to build a general conceptual understanding of what shapes chaos within microbial communities? If so, again I’d expect to see a far broader exploration of different microbial systems – ie different numbers of species, different interaction types etc etc. Even just within the 3 species, 2QS model presented, I’d want to know whether the the parameter relationships seen in m850 hold for other model formulations. I would also want more of a discussion of the biological relevance, ie how do the patterns observed compare to what we see in nature, or in existing (in vitro) synthetic communities?

c. Or, was the goal to really dig into the behavior of m850? If so, here I’d definitely expect to see some comparison against empirical data – for example experiments altering dilution rates and showing transitions to chaos.

Smaller comments

Pg 2 “While these abstractions are suitable in some circumstances, using them to inform gene regulation networks and community design can be difficult” – why?

2.2 “Here we see some evidence for a peak in…” this seems like an interesting result relative to past work, picking up the theme of how generalizable are your results – would have been nice to explore further.

Para 1 page 7, this section wasn’t super clear, perhaps needs stepping through a bit how these results compare to previous literature.

Discussion “We show that we can expect to find chaotic states in relatively small synthetic microbial systems, which has important ramifications for the field” Coming back to my concerns about novelty, we already know you can find chaos in 3 species systems, I’m not sure the distinction between natural vs synthetic is super meaningful, so where really is the novelty?

“Although chaotic attractors are generally thought to be sparse in low dimensional systems, we have shown their existence in realistic synthetic microbial systems.” Are they not also sparse in your system 25/4000something? Or are they far more sparse in your system than others?

Reviewer #2: The authors develop and apply methods for for identifying chaotic parameter regimes using ABC SMC, and thereby identifying models/model topologies that permit chaotic behaviour. This is applied in the context of models of small microbial communities governed by resource competition, intercellular communication and competitive bacteriocin interactions. The inference and "search for desired behaviour" aspects seem to build upon the corresponding author's 2011 PNAS paper (an observation that is not intended as a criticism, and I would stress that the present manuscript is certainly not lacking in novelty). I found the manuscript to be interesting and generally clearly written (and the Zenodo repo is also appreciated), and have included a few points and suggestions below that I hope the authors may find useful.

Minor points:

0. There is an additional key paper that I think the authors should cite, which I suspect has been inadvertently omitted (unless I missed it): Barnes, C. P., Silk, D., Sheng, X., & Stumpf, M. P. (2011). Bayesian design of synthetic biological systems. Proceedings of the National Academy of Sciences, 108(37), 15190-15195.

1. The abstract could be more specific, which may help to encourage a larger readership for the manuscript. In particular, a brief description of the "methodology to explore the potential for chaotic dynamics" would be appreciated, mentioning ABC and the "filtering" approach of looking for oscillations and then chaos.

2. General comment: when talking about "posterior probabilities", I think it needs to be made clear that these are posterior probabilities *given* the desired/target qualitative behaviour (to distinguish from the more usual situation where they would be conditioned on an observed dataset). Perhaps this need only be done once, the first time the expression is used, to make clear what is meant.

3. Caption for Fig 1: Spell out what A_i, B_j and N_k denote in panel (a). In the second and third boxes of panel (a), could the criteria for oscillation and chaos be briefly summarised, e.g. for oscillations, even just listing the three d_{oi} and then referring forward in the caption to the Methods, would help to guide the reader. For (c), I'm not sure the y-axis "OD 600" is clearly explained, and I was unclear what was meant in the caption by "An example time series representative of the dataset" (which dataset?). Finally, I don't think the figure at the bottom panel c is explicitly described in the caption or text?

4. Is Algorithm 2 (ABC SMC) referenced in the text? Apologies if I missed it. In this algorithm there is a single distance, d (line 15 of algorithm), whereas for the "Oscillatory population dynamic objective" there are three distances. How does d relate to these three distances?

5. Caption for Fig 2: "The bar chart shows the mean model posterior probability across three experiments, represented by the scatter points..." Apologies, I may be missing something here: to what does "the scatter points" refer?

6. Near start of Section 2.2, stating that m_k denotes the k-th considered model would aid the reader.

7. Prior model probabilities: I'm not sure if these were stated anywhere (apologies if I missed it). Presumably uniform over all considered models?

8. "The classic debate on the complexity-stability relationship in theoretical ecology is likely highly dependent on the nature of the biological interactions involved". This is an interesting point, which several authors have considered, e.g. (i) Allesina, S., & Tang, S. (2012). Stability criteria for complex ecosystems. Nature, 483(7388), 205-208; and (ii) Kirk, P., Rolando, D. M., MacLean, A. L., & Stumpf, M. P. (2015). Conditional random matrix ensembles and the stability of dynamical systems. New Journal of Physics, 17(8), 083025.

Do the authors believe that some of the "design principles" for stability (e.g. the stability criteria described by Allesina and Tang) could provide helpful prior information, e.g. to help restrict the set of models that is considered?

9. In Section 2.3, when the random forest classifier is introduced, it took me a little while to understand what was being done. I appreciate that the details are provided in the Methods, but I felt that perhaps the definition of the training and test sets needed to be briefly provided in Section 2.3, to aid comprehension. It would moreover seem appropriate to cite Breiman. Perhaps also some of the ABC-RF work (e.g. Pudlo, P., Marin, J. M., Estoup, A., Cornuet, J. M., Gautier, M., & Robert, C. P., 2016), if the authors feel it would be appropriate.

10. Given the definition of chaos, it might be worth briefly commenting on the limitations of fixed precision ODE solvers, such as the one used here. This is not an area about which I know a great deal, but I would direct the authors toward, for example: Hu, T., & Liao, S. (2020). On the risks of using double precision in numerical simulations of spatio-temporal chaos. Journal of Computational Physics, 418. Exploring this in detail would be beyond the scope of the current manuscript, but perhaps a comment to say that this could be worthy of further investigation (if the authors agree that it might be?).

11. Figure 3d -- the green dots are a little difficult to see. Could a different marker shape be used? Figure 3c -- include units (bits) on the y-axis.

12. I was unable to run the Matlab p files provided in the repo, receiving an error: "Invalid pfile. The file header is corrupt." This may just be my system (I did not have chance to look into the error in any depth), but I would be very grateful if the authors could look into this (even if the conclusion of their investigation is that this is a problem that is specific to this reviewer, in which case they can just throw this back to me to investigate).

Typographical and grammatical points:

a. Intro, first para "due fact"

b. Intro, 2nd para: "as the number of dimensions in the network grow". grow  grows

c. p4: "Due to the limited time frame from which calculate"

d. p4: "m850 contains four expressed parts and possess". possess  possesses

e. p9 " appears to more"

f. p11: "It’s important" [avoid contraction]

g. p14, 4.1 "includes includes"

h. p14: "Each strain is defined by it’s sensitivities". it's  its

i. p19: "Simulations in which an strain"

j. p24 (bottom): "as a functions of S"; also "increases as function of S"

**Have the authors made all data and (if applicable) computational code underlying the findings in their manuscript fully available?**

Reviewer #1: Yes

Reviewer #2: Yes

PLOS authors have the option to publish the peer review history of their article (what does this mean?). If published, this will include your full peer review and any attached files.

Reviewer #1: No

Reviewer #2: No
---

## [Decision Letter · Decision Letter 1]

29 Jul 2022

Dear Dr Barnes,

Thank you very much for submitting your manuscript "Chaos in synthetic microbial communities" for consideration at PLOS Computational Biology. As with all papers reviewed by the journal, your manuscript was reviewed by members of the editorial board and by several independent reviewers. The reviewers appreciated the attention to an important topic. Based on the reviews, we are likely to accept this manuscript for publication, providing that you modify the manuscript according to the review recommendations.

The suggestions are minor and should be easy to address. Assuming they are adequately addressed, we do not see the need for a further round of external review.

Sincerely,

Christopher Rao

Associate Editor

PLOS Computational Biology

Jason Papin

Editor-in-Chief

PLOS Computational Biology

[LINK]

Reviewer's Responses to Questions

**Comments to the Authors:**

Reviewer #1: The authors have adequately addressed all of my previous comments. I particularly appreciate the inclusion of the 4 species gLV model which helps with both the generality and the understanding of their method.

Small comments:

I am still curious about the peak in probability of chaos with increasing number of parts (line 127) - is there any way of determining whether this is real and not an erroneous prediction of your model (eg unbiasedly sampling from models with varying number of parts and comparing their stability within simulations?)

More importantly I do not think the comparison to the complexity-stability debate is entirely appropriate here, given this typically refers to complexity in terms of number of species or number of interactions, not complexity of the interactions themselves. I would rephrase this section to make this distinction much clearer.

Reviewer #2: Thanks to the authors for addressing my points. I have only very minor comments remaining:

1. I still think it would be appropriate to cite Breiman re: random forests; namely, Breiman, L., 2001. Random forests. Machine learning, 45(1), pp.5-32.

2. I think the figure numbers in the README of the Zenodo repo may now be out of sync with the manuscript (but I can understand the authors not wishing to update these until the end of the review process, to avoid having to repeatedly update the repo).

**Have the authors made all data and (if applicable) computational code underlying the findings in their manuscript fully available?**

Reviewer #1: Yes

Reviewer #2: Yes

PLOS authors have the option to publish the peer review history of their article (what does this mean?). If published, this will include your full peer review and any attached files.

Reviewer #1: No

Reviewer #2: No

Figure Files:

Data Requirements:

Reproducibility:

References:

---

## [Editor Report · Decision Letter 2]

7 Sep 2022

Dear Dr Barnes,

We are pleased to inform you that your manuscript 'Chaos in synthetic microbial communities' has been provisionally accepted for publication in PLOS Computational Biology.

Best regards,

Christopher Rao

Academic Editor

PLOS Computational Biology

Jason Papin

Editor-in-Chief

PLOS Computational Biology

---

## [Editor Report · Acceptance letter]

4 Oct 2022

PCOMPBIOL-D-21-02225R2 

Chaos in synthetic microbial communities

Dear Dr Barnes,

I am pleased to inform you that your manuscript has been formally accepted for publication in PLOS Computational Biology. Your manuscript is now with our production department and you will be notified of the publication date in due course.

With kind regards,

Olena Szabo
